

# Solubilization of skin collagen improves the accuracy and reliability of stable isotope measurements

Alexandra A.Y. Derian[1], Ryan Pawlowski[2] and Paul Szpak[2]

[1] Environmental and Life Sciences Graduate Program, Trent University, Peterborough, Ontario, Canada
[2] Anthropology, Trent University, Peterborough, Ontario, Canada

## ABSTRACT

Stable isotope analysis of skin collagen is useful for detecting short-term or seasonal diet. Preparation of skin for stable isotope analysis varies across laboratories, and this may impact the comparability of data. It is important to understand the effects of different preparatory protocols on the stable isotopic and elemental compositions of skin samples. Using a Eurovector 3,300 elemental analyzer coupled to a Nu Horizon isotope ratio mass spectrometer, we tested the impact of three treatment variants (refluxing at three temperatures to remove non-collagenous proteins, treatment with sodium hydroxide (NaOH), and chemical lipid extraction using 2:1 chloroform:methanol) on the stable isotope ($\delta^{13}$carbon (C) and $\delta^{15}$nitrogen (N)) and elemental (wt% C, and wt% N) composition of pig (*Sus scrofa domesticus*) skin. The refluxing step produced pig skin with higher $\delta^{13}$C values, lower C:N$_{Atomic}$ ratios, less variable C:N$_{Atomic}$ ratios, wt% C, and wt% N. The chemical lipid extraction also produced higher, more reliable $\delta^{13}$C values and lower, less variable C:N$_{Atomic}$ ratios. The isotopic data in the lipid-extracted and refluxed samples were more consistent in the refluxed samples than the non-refluxed and non-lipid-extracted samples, as determined by the elemental compositions.

## INTRODUCTION

Stable isotope analysis in ecology and archaeology emerged in the 1970s, and a range of bodily tissues such as bone (collagen and bioapatite), teeth (enamel and dentine), hair, muscle, and skin are now widely used for the reconstruction of a consumer's diet and mobility (*Britton, 2017*; *Fry, 2006*; *Peterson & Fry, 1987*; *Tieszen et al., 1983*; *O'Connell, 2023*). Hair and teeth (both enamel and dentine collagen) are inert tissues once formed, and thus reflect the diet of an individual during the period of tissue formation (*Tieszen et al., 1983*; *Thackeray & Lee-Thorp, 1992*; *Macko et al., 1999*). Bone collagen is believed to represent the average diet of an individual over several years, but recent studies suggest that it is more like a mosaic, with different regions of bone potentially reflecting different periods of life (*Matsubayashi & Tayasu, 2019*; *Hall et al., Forthcoming 2025*) Muscle turnover is not well-understood, but approximately reflects a period of months prior to sampling (*Tieszen et al., 1983*).

Corresponding author
Alexandra A.Y. Derian,
alexderian@trentu.ca

The isotopic composition of skin collagen reflects the months preceding tissue sampling or death (although, the rate isotopic turnover in skin is species-dependent and not well studied) and is therefore useful for detecting recent or seasonal changes in diet when compared with tissues that have slower turnover rates (*Alves-Stanley & Worthy, 2009*; *Tieszen et al., 1983*; *Giménez et al., 2016*; *White & Schwarcz, 1994*). There are three layers in mammalian skin: the epidermis, dermis, and hypodermis (*Akat et al., 2022*). The epidermis is primarily composed of keratinocytes, which synthesize keratin, as well as melanocytes, Merkel cells, and Langerhans cells (*Akat et al., 2022*; *Watt, 2014*). The dermis is abundant in fibroblasts and fibrillar collagen (*Watt, 2014*). The hypodermis (subcutaneous fat layer) is composed of white adipocytes (*Akat et al., 2022*; *Watt, 2014*). The most abundant proteins in the dermis, which is the thickest layer of skin, are collagen (approximately 70–80% by dry weight), elastin (2–4%), and glycosaminoglycans (0.1–0.3%). Lipids are present in all layers of skin, though the approximate percentage of lipids by weight in skin is uncertain, and variable depending on species and age (*Knox & O'Boyle, 2021*; *Waller & Maibach, 2006*; *Zwara, Wertheim-Tysarowska & Mika, 2021*). Although modest differences in skin (*e.g.*, epidermal thickness, pigmentation, type of immune cells) exist across mammalian taxa, the overall structure and chemical composition is consistent (*Akat et al., 2022*). Therefore, while the present study uses pig (*Sus scrofa domesticus*) skin to test different preparatory protocols for stable isotope analysis, the experiment and results should be applicable to non-porcine mammalian skin.

The effects of different pre-treatment procedures on the elemental and isotopic compositions of skin have been under-researched in comparison to other tissues, such as bone and tooth dentine collagen (*Longin, 1971*; *Brown et al., 1988*; *De Niro, 1985*; *Wilson & Szpak, 2022a*; *Wilson & Szpak, 2022b*; *Jørkov, Heinemeier & Lynnerup, 2007*; *Ambrose, 1990*; *van Klinken, 1999*; *van Klinken & Hedges, 1995*). Although previous studies have tested the isotopic homogeneity of skin (*Todd et al., 2009*), the effects of removing lipids on elemental carbon and $\delta^{13}$C values (*Ryan et al., 2012*; *Todd et al., 1997*), and the removal of preservatives from mummified skin (*Cockitt, Lamb & Metcalfe, 2020*) and parchment (*Brock, 2013*; *Doherty et al., 2021*; *Davis et al., 2024*) there has not been a systematic test of sample preparation methods of fresh skin collagen. A commonly employed technique in the preparation of collagen from bones and teeth involves 'refluxing' or solubilizing collagen through heating the sample in weakly acidic solution (*Longin, 1971*; *Brown et al., 1988*). Refluxing of bone collagen results in a homogeneous and pure analyte that very closely matches the theoretical elemental composition of collagen based on its amino acid composition (*Wilson & Szpak, 2022a*; *Guiry & Szpak, 2020*). Moreover, the collagen is shelf stable with respect to its isotopic compositions for at least ten years when stored at room temperature in sealed vials. We therefore investigated whether preparing skin collagen by refluxing produced a superior analyte relative to whole skin. We evaluated the quality of refluxed and non-refluxed pig skin samples by comparing their carbon and nitrogen elemental and isotopic ($\delta^{13}$C, $\delta^{15}$N) compositions.

Most of the published studies presenting stable isotope compositions for skin have examined marine fauna such as cetaceans, turtles, and pinnipeds (*Todd et al., 2009*; *Ryan et al., 2012*; *Todd et al., 1997*; *Borrell et al., 2018*; *Gelippi et al., 2023*; *Kiszka et al., 2010*;

*Turner Tomaszewicz et al., 2017*; *Vanderklift et al., 2020*). Prior to analysis, skin samples are typically lyophilized or oven-dried, homogenized to a powder, and treated with a solvent (such as chloroform:methanol, dichloromethane:methanol, and *n*-hexane:acetone) to remove lipids (*Todd et al., 2009*; *Ryan et al., 2012*; *Todd et al., 1997*; *Borrell et al., 2018*; *Miles, Gibbon & Hayden, 2025*). Skin appears to be isotopically homogeneous (*Borrell et al., 2018*; *Arregui et al., 2017*), but the proportion of lipids in the skin can vary by location on the body (*Todd et al., 2009*; *Ryan et al., 2012*; *Todd et al., 1997*). The removal of lipids prior to analysis should therefore result in more accurate $\delta^{13}$C values and elemental compositions closer to pure collagen relative to mathematical corrections based on carbon:nitrogen (C:N) ratios (*Ryan et al., 2012*).

Other studies of preparation of skin for stable isotope analysis have been conducted on mummified human remains and parchment. Ancient skin is rarely preserved naturally, and in most of the studies involving archaeological skin, the primary concern is removing preservative solvents that may affect isotopic and elemental compositions of the samples (*Cockitt, Lamb & Metcalfe, 2020*; *Brock, 2013*; *Doherty et al., 2021*; *Davis et al., 2024*). Although the samples included in our study are modern and not treated with preservative solvents, some procedures used to prepare collagen from archaeological skin are relevant to our study. Mummified skin and parchment samples are often treated with solutions such as chloroform, methanol, dichloromethane, and toluene to remove lipids prior to isotopic analysis (*Cockitt, Lamb & Metcalfe, 2020*; *Doherty et al., 2021*; *Finucane, 2007*; *Hyland, Millaire & Szpak, 2021*). Some studies refluxed samples to extract collagen (*Hyland, Millaire & Szpak, 2021*), often including an ultrafiltration step following solubilization to isolate only high molecular weight compounds (*Cockitt, Lamb & Metcalfe, 2020*; *Brock, 2013*; *Doherty et al., 2021*; *Finucane, 2007*). The temperature used to reflux samples in these studies ranged from 60–80 °C and length of refluxing time ranged from 10–48 h (*Cockitt, Lamb & Metcalfe, 2020*; *Brock, 2013*; *Doherty et al., 2021*; *Finucane, 2007*; *Hyland, Millaire & Szpak, 2021*). *Doherty et al. (2021)*, *Doherty et al. (2022)* also included an acidification step prior to refluxing, in which they treated both parchment and unprocessed skin samples with 0.6 M hydrochloric acid (HCl) for 1 and 6 h, respectively. *Brock (2013)* compared various acid–base-acid treatments, one of which was a modified bone collagen extraction procedure—parchment was treated with 0.5 M HCl followed by 0.1 M sodium hydroxide (NaOH), followed by 0.5 M HCl prior to refluxing. *Davis et al. (2024)* also tested acid–base-acid treatments, and a modified bone collagen extraction procedure on leather, parchment, and dried but untreated rabbit skin from AD 1985–1990 for $^{14}$C dating. Their procedure used pretreatments with 0.1 M HCl at 4 °C for 16 h, at least one wash with 0.1 M NaOH at 20 °C for 10 min, and one wash with 0.1 M HCl at 20 °C for 10 min. Samples were then refluxed in 0.01 M HCl at 60 °C for 10 h. Refluxing was found to lower C:N$_{\text{Atomic}}$ ratios closer to the expected ratio for pure collagen, and improve the accuracy of $^{14}$C dates.

The variability in pretreatment procedures across laboratories means that stable isotope measurements of skin could be differentially influenced by the presence of non-collagenous materials (*i.e.,* proteins and/or lipids), thus affecting the comparability of data across studies. Moreover, the widespread use of protocols that produce heterogeneous material will add considerable noise (isotopic variability) that does not reflect the life history of the sampled

organisms. Non-collagenous materials can have significant effects on both a sample's stable isotope ratios, which are used to make inferences about a consumer's life history, and the elemental compositions, which are used to detect sample contamination or degradation (*Brown et al., 1988*; *Guiry & Szpak, 2020*; *Trayler, Landa & Kim, 2023*). Removal of lipids, exogenous contaminants (*e.g.*, humic acids), bioapatite, and non-collagenous proteins (NCPs) with chemical pretreatments is common in the preparation of collagen from bone for isotopic analysis to produce the purest possible analyte (*Longin, 1971*; *Brown et al., 1988*; *De Niro, 1985*; *Wilson & Szpak, 2022a*; *van Klinken, 1999*; *Guiry & Szpak, 2020*; *Guiry, Szpak & Richards, 2016*; *Guiry & Szpak, 2021*). The primary objective of the current study was to test whether isolating collagen from skin improved the accuracy and reliability of stable carbon and nitrogen isotope measurements, compared with analyzing whole, untreated skin. Providing a better understanding of the effects of pretreatment procedures (or lack thereof) on stable isotopic and elemental compositions of skin/skin collagen has implications for ensuring that stable isotopic data accurately reflect the diet of a consumer, and for improving the comparability of data across laboratories.

## METHODS

### Sample selection

Three domestic pig (*Sus scrofa domesticus*) distal limbs (termed Pig 1, Pig 2, and Pig 3 here) were obtained from a grocery store in Canada and sampled in this study. Pig 1 is a right manus (distal forelimb), Pig 2 is a left manus, and Pig 3 is a right pes (distal hindlimb). The distal epiphyses of the metapodials were fused, indicating that the individual(s) was/were at least two years of age (*Silver, 1963*). The three distal limbs could belong to the same individual, but we do not have evidence to make that determination. No life history or health information for the individual(s) is available. The limbs were stored in a freezer at −20 °C and thawed at room temperature prior to sampling. Using a scalpel, we removed chunks of skin that included the epidermis and dermis layers. The initial mass of the individual skin samples ranged from 45.3–516.1 mg. We also sampled ~300 mg of trabecular bone from a carpal/tarsal of each distal limb for comparison of the isotopic and elemental compositions of the skin.

### Sample treatments

For each temperature treatment, 10 skin samples were collected from each distal limb. This resulted in a total of 30 samples of refluxed collagen, and 30 insoluble skin residue (the material that did not dissolve into solution during the refluxing step) per treatment. Five skin samples from each distal limb were collected for NaOH treatment testing, generating 15 samples of refluxed collagen and insoluble skin, respectively. We collected fewer samples for this treatment because the primary purpose of the study was to test the effects of refluxing on skin collagen stable isotope measurements rather than the effects of different chemical pretreatments. Additionally, 10 skin samples from each distal limb were collected and treated to remove lipids, but were not refluxed (0/LE), and 10 were not pretreated prior to analysis (0/0), resulting in a total of 30 0/LE and 30 0/0 skin samples. The treatments for all skin samples are summarized in Table 1 and Fig. 1.

**Table 1  Pretreatments of samples included in this study.**

| Treatment | Material | Chemical pretreatment | Refluxing solution | Refluxing temperature (°C) | $n$ |
|---|---|---|---|---|---|
| 65/LE | Skin | Chloroform: methanol | 0.1 M HCl | 65 | 30 |
| 62/LE | Skin | Chloroform: methanol | 0.1 M HCl | 62 | 30 |
| 58/LE | Skin | Chloroform: methanol | 0.1 M HCl | 58 | 30 |
| 62/LE*A | Skin | Chloroform: methanol; NaOH | 0.1 M HCl | 62 | 30 |
| 0/LE | Skin | Chloroform: methanol | N/A | N/A | 30 |
| 0/0 | Skin | None | N/A | N/A | 30 |
| B62/LE | Bone | Chloroform: methanol | 0.1 M HCl | 62 | 3 |

To extract lipids, we added eight mL of 2:1 chloroform:methanol to each sample in a 16 × 125 mm 19 mL borosilicate glass culture tube. The samples were sonicated for 3 h, and the solution was refreshed every 60 min. Samples in one of the treatments (62/LE/NaOH) were treated with 0.1 M NaOH for 30 min and rinsed to neutrality with Type I water. We included this step to test whether treatment with NaOH would alter the elemental compositions of the extracts by removing keratin from the sample, as NaOH is known to effectively degrade keratin disulfide bonds (*Shavandi et al., 2017*; *Zhang et al., 2015*). Following the lipid extraction procedure, 3.5 mL of 0.01 M HCl was added to the skin samples. All samples that were subjected to a refluxing step were refluxed for 36 h at 58, 62, or 65 °C. These temperatures were chosen because they spanned the range identified by *Brown et al. (1988)* to maximize collagen yields from bone while minimizing collagen degradation.

Following refluxing of the skin samples, the weak acid solution containing the collagen was collected using a clean glass Pasteur pipette, leaving the insoluble skin reside in the tube. Both the collagen solution and the insoluble residue were transferred to clean four mL glass vials, frozen at −20 °C for at least 24 h, and lyophilized for 48 h. We used neither filters to exclude larger particulate matter nor centrifugal ultrafilters to remove low molecular weight compounds in the preparation of refluxed collagen. The latter were found by *Guiry, Szpak & Richards (2016)* to be less effective at removing lipids than 2:1 chloroform:methanol (and may have selectively *retained* lipids), they reduced collagen yields, and altered amino acid sequences. Filters to exclude large particulate matter were not necessary as centrifugation of the refluxed collagen solution made it easy to separate this material. The non-refluxed skin (0/LE) and non-lipid extracted/non-refluxed skin (0/0) were frozen and lyophilized.

Bone samples (B62/LE) were cut using a rotary tool fitted with a diamond-tipped wheel. Lipids were extracted as described above. After drying in a fume hood for 24 h, nine mL of 0.5 M HCl was added to the bone samples, which were left on an orbital shaker at room temperature for 2 d. After being rinsed to neutrality with Type I water, 3.5 mL of 0.01 M HCl was added to the bone samples, which were then refluxed at 62 °C for 36 h. The refluxed collagen solution was transferred to clean four mL glass vials, frozen at −20 °C for at least 24 h, and lyophilized for 48 h.

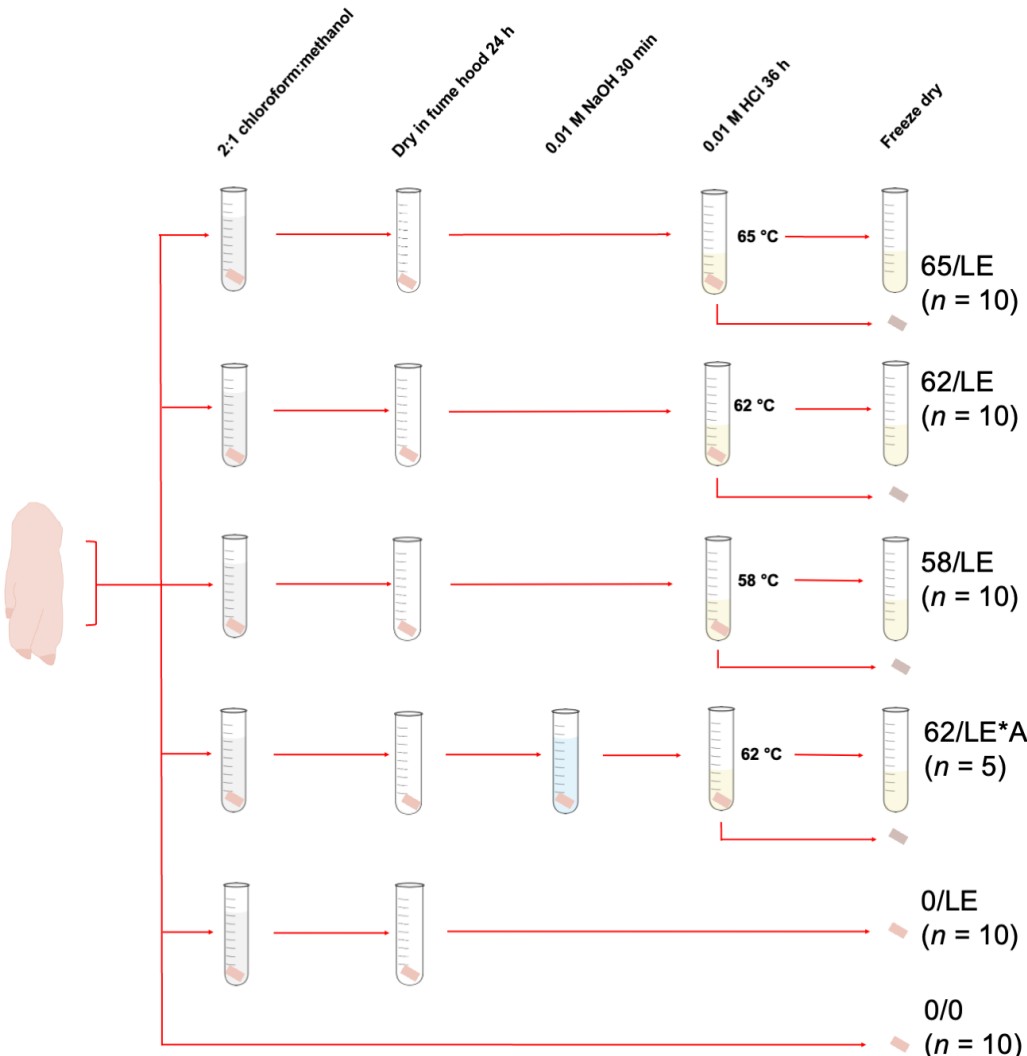

**Figure 1  Pretreatments of pig skin included in this study.**

## Sample analysis

$550 \pm 50$ μm of collagen, skin, or insoluble residue was weighed into tin capsules, and the elemental and isotopic compositions were determined with a Eurovector 3300 Elemental Analyzer (EA) coupled to a Nu Horizon Isotope Ratio Mass Spectrometer (IRMS) in the Water Quality Centre at Trent University.

Stable carbon and nitrogen isotopic compositions were calibrated relative to the Vienna Pee Dee Belemnite (VPBD) and Atmospheric Inhalable Reservoir (AIR) scales using IAEA-CH-6 (sucrose, $\delta^{13}C = -10.45 \pm 0.03‰$), IAEA-CH-7 (polyethylene, $\delta^{13}C = -32.15 \pm 0.05‰$), USGS25 (ammonium sulfate, $\delta^{15}N = -30.41 \pm 0.27‰$), USGS40 (glutamic acid, $\delta^{13}C = -26.39 \pm 0.04‰$, $\delta^{15}N = -4.52 \pm 0.06‰$) USGS41a (glutamic acid, $\delta^{13}C = 36.55 \pm 0.08‰$, $\delta^{15}N = 47.55 \pm 0.15‰$), USGS62 (caffeine, $\delta^{13}C = -14.79 \pm 0.04‰$, $\delta^{15}N = 20.17 \pm 0.06‰$), and USGS63 (caffeine, $\delta^{13}C = -1.17 \pm 0.04‰$, $\delta^{15}N = 37.83$

$\pm$ 0.06‰. Six in-house standards, matrix-matched to mammalian skin (*i.e.,* having similar elemental compositions to collagen) were used to monitor measurement uncertainty: SRM-1 (caribou bone collagen, $\delta^{13}C = -19.40 \pm 0.08$‰, $\delta^{15}N = 1.88 \pm 0.14$‰), SRM-4 (wheat gluten, $\delta^{13}C = -26.80 \pm 0.12$‰, $\delta^{15}N = 5.22 \pm 0.15$‰), SRM-14 (polar bear bone collagen, $\delta^{13}C = -13.62 \pm 0.07$‰, $\delta^{15}N = 21.50 \pm 0.17$‰), SRM-26 (commercial marine collagen, $\delta^{13}C = -16.9 \pm 0.1$‰, $\delta^{15}N = 14.71 \pm 0.18$‰), and SRM-28 (alanine $\delta^{13}C = -16.27 \pm 0.1$‰, $\delta^{15}N = -1.92 \pm 0.17$‰). Precision, or *u(Rw)* was calculated to be $\pm$ 0.17‰ for $\delta^{13}C$ and $\pm$ 0.39‰ for $\delta^{15}N$, based on repeated measurements of calibration standards, check standards, and sample replicates. The systematic error, or *u(bias)* was calculated to be $\pm$ 0.12‰ for $\delta^{13}C$ and $\pm$ 0.19‰ for $\delta^{15}N$ by comparing the observed compared to the known values of the standards. The overall analytical uncertainty was determined to be $\pm$ 0.21‰ for $\delta^{13}C$ and $\pm$ 0.39‰ for $\delta^{15}N$ (*Szpak, Metcalfe & Macdonald, 2017*). Note that the total analytical uncertainty estimated for $\delta^{15}N$ was slightly higher than what might be typically reported due to heterogeneity observed within analyzed duplicates of 0/0 and 0/LE treatments.

## Statistical analysis

Whenever possible, we combined the results from the three distal limbs however, different diets and physiological stressors in life could impact the isotopic compositions of skin collagen. Therefore, we combined data from the distal limbs for elemental data (wt% C, wt% N, C:N$_{Atomic}$) since the amino acid composition of the skin protein should not vary significantly with diet or physiology, but separated the isotopic data by distal limb for statistical comparisons. To minimize the impact of any dietary or physiological differences among pigs on the isotopic compositions, rather than plotting absolute $\delta^{13}C$ or $\delta^{15}N$ values, we plotted the difference in $\delta^{13}C$ or $\delta^{15}N$ from the median value for a given pig within a given treatment.

A Shapiro–Wilk test was used to assess the normality of isotopic and elemental compositions for the different treatments. For normally distributed samples, we used Student's *t*-test (for samples with equal variances), Welch's *t*-test (for samples with unequal variances), and one-way analysis of variance (ANOVA) with Tukey's Pairwise *post-hoc* test. For non-normally distributed samples, we used Mann–Whitney *U* test, and Kruskal–Wallis test with Dunn's *post-hoc* test. To evaluate the overall isotopic variance across treatments, we calculated the combined standard deviations of $\delta^{13}C$ and $\delta^{15}N$ for the different treatments used on skin samples.

In the context of elemental compositions, we discuss atomic C:N ratios (C:N$_{atomic}$), which we define as the ratio of carbon to nitrogen atoms in a compound, such that caffeine ($C_8H_{10}N_4O_2$) has a C:N$_{atomic}$ of 8:4 or 2. We differentiate the C:N$_{atomic}$ from the molecular ratio of carbon to nitrogen (C:N$_{molecular}$), which for caffeine would be 1.71 or $(8 \times 12.011) \times (4 \times 14.007)^{-1}$[34].

## RESULTS

All individual isotopic and elemental compositions are presented in full in Table S1 along with relevant contextual data, and are visualized in Figs. S1–S3. Refluxing significantly

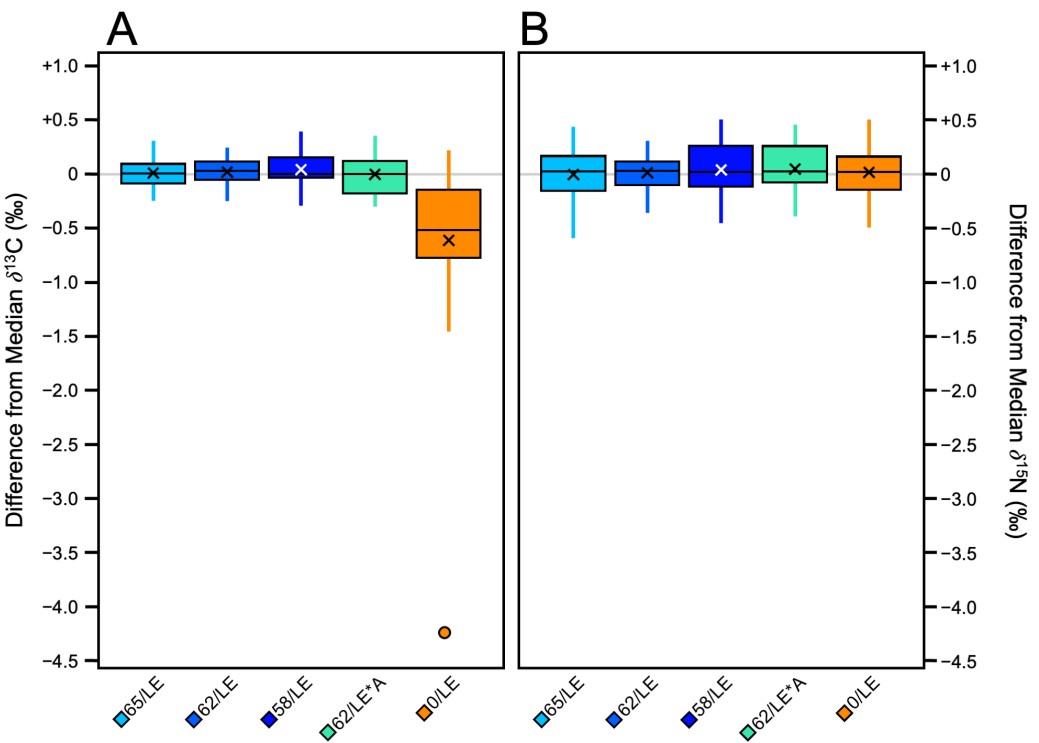

**Figure 2** **Boxplots visualizing per mille difference from distal limb median.** (A) $\delta^{13}$C. (B) $\delta^{15}$N.

affected the isotopic and elemental compositions of the skin samples. Within each treatment, the $\delta^{13}$C, and $\delta^{15}$N values of skin samples were within <1‰ of one another (Table S2), which corroborates the findings of previous studies that skin appears to be isotopically homogeneous (*Borrell et al., 2018*; *Arregui et al., 2017*). We therefore interpreted differences in the isotopic and elemental compositions of the skin samples within each distal limb to result from the pretreatment procedures rather than changes in diet or physiology of the individual pigs.

## Effects of the treatments on $\delta^{13}$C and $\delta^{15}$N values

The refluxed skin samples had significantly different $\delta^{13}$C values than the non-refluxed skin in all three distal limbs (Pig 1: $H$ [4] = 32.39, $p$ = 1.576E–6, $\varepsilon^2$ = 0.74); Pig 2: Pig 2 $H$ [4] = 26.94, $p$ = 2.027E–05, $\varepsilon^2$ = 0.61; Pig 3: $H$ [4] = 26.73, $p$ = 2.219E–05, $\varepsilon^2$ = 0.61; Table S3). The non-refluxed samples (0/LE) displayed greater variability in $\delta^{13}$C than the refluxed skin in all three distal limbs (Fig. 2A), indicating that the refluxing the skin produced more consistent $\delta^{13}$C values.

There were significant differences in $\delta^{15}$N values among the treatments in Pig 2 ($F$ [4, 40] = 3.85, $p$ = 0.009, $\eta_p^2$ = 0.28) and Pig 3 ($F$ [4, 40] = 4.18, $p$ = 0.006, $\eta_p^2$ = 0.29). There were no significant differences in $\delta^{15}$N values among the treatments in Pig 1. Unlike $\delta^{13}$C, there was no clear relationship between treatment and the variability of the $\delta^{15}$N values, and the magnitude of variability from the median of each distal limb was much smaller

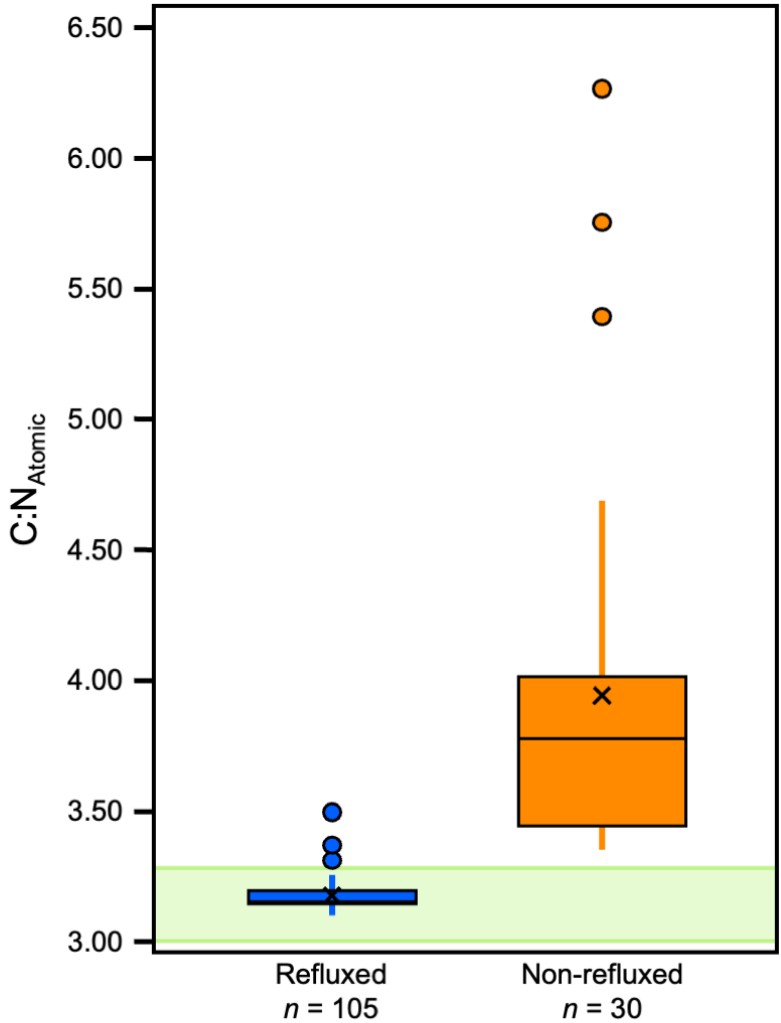

**Figure 3** Boxplots displaying C:N$_{Atomic}$ distribution within refluxed and non-refluxed samples.

(Fig. 2B). Tukey's *post-hoc* tests revealed which treatments were significantly different from one another (Table S4).

### Effects of the treatments on atomic C:N ratios

The most pronounced difference between refluxed and non-refluxed samples was in the atomic C:N ratios (Table S5; Fig. 3). Refluxed skin collagen (all treatments combined) had atomic C:N ratios that were significantly lower than the non-refluxed whole skin (0/LE; $z = 8.25$, $p = 1.5972E{-}16$, $r = 0.71$). The refluxed skin atomic C:N values also displayed much lower variance than the non-refluxed skin ($F = 121.73$, $p = 1.3551E{-}67$, $f^2 = 2.27$; Fig. 3).

Significant differences were also found between specific, individual treatments ($H$ [4] $= 87.62$, $p = 3.186E{-}18$, $\varepsilon^2 = 0.65$). There were significant differences in atomic C:N ratios between 65/LE and 62/LE, 62/LE and 58/LE, and 62/LE and 62/LE*A, suggesting

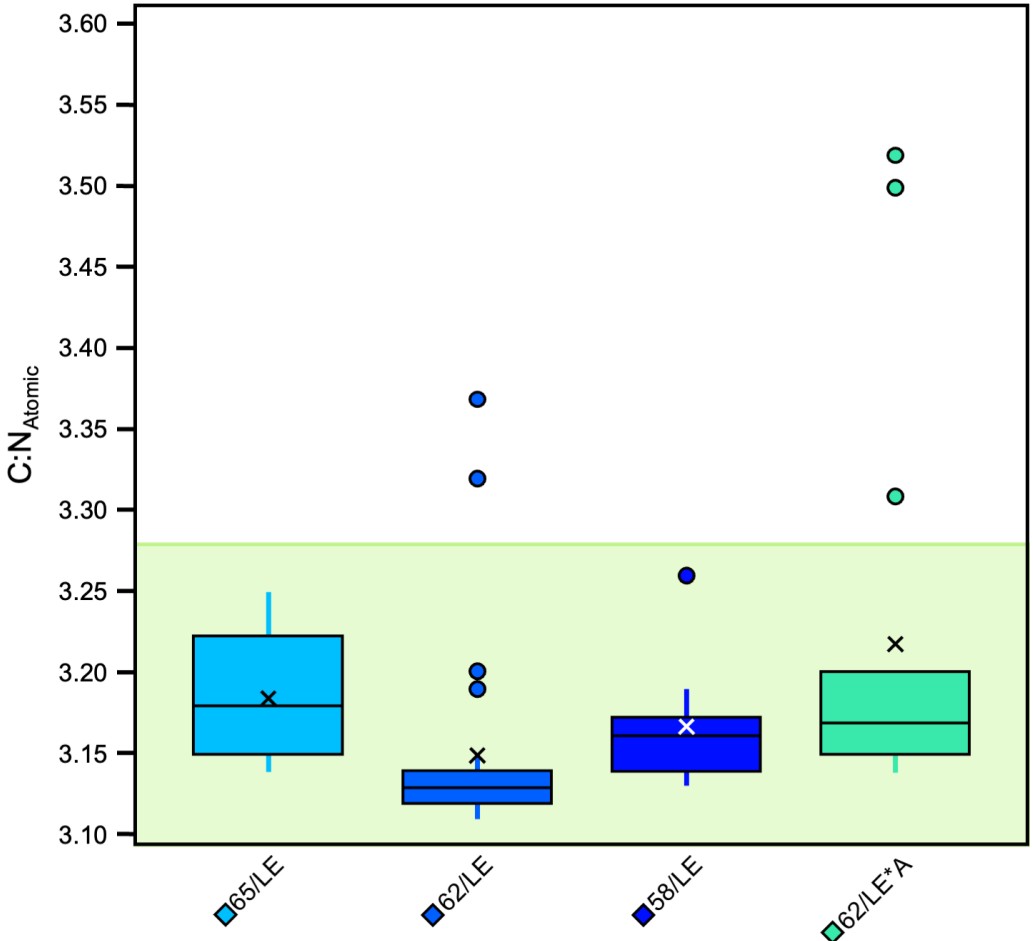

**Figure 4** Boxplot visualizing C:N$_{Atomic}$ across treatments.

that refluxing treatments may systematically produce different elemental compositions (Table S6). The atomic C:N ratios of all refluxing treatments were each found to be significantly different from the 0/LE treatment. 62/LE had the smallest range of atomic C:N ratios, while 65/LE was the only treatment to feature no outlier with all samples displaying atomic C:N ratios that are consistent with mammalian collagen (Fig. 4). Mann–Whitney U (Pig 1 and 2) and Welch's $t$-test (Pig 3) determined that samples with atomic C:N ratios above 3.28 had significantly different $\delta^{13}$C values than those with atomic C:N ratios below 3.28 (Pig 1: $z = 4.77$, $p = 1.8801\text{E}{-}06$, $r = 0.71$; Pig 2: $z = 4.77$, $p = 1.8816\text{E}{-}06$, $r = 0.71$; Pig 3: $t\,(43) = 5.37$, $p = 7.2337\text{E}{-}05$, $d = 2.27$; Fig. 5). The value of 3.28 for C:N$_{atomic}$ is the cutoff for mammalian bone collagen samples suspected to be lipid-contaminated (*Guiry & Szpak, 2020*).

## Effects of the treatments on wt% C and wt% N

The non-refluxed samples displayed the largest range and standard deviation of wt% C (Table S2). 65/LE displayed the lowest standard deviation of wt % C. Apart from the

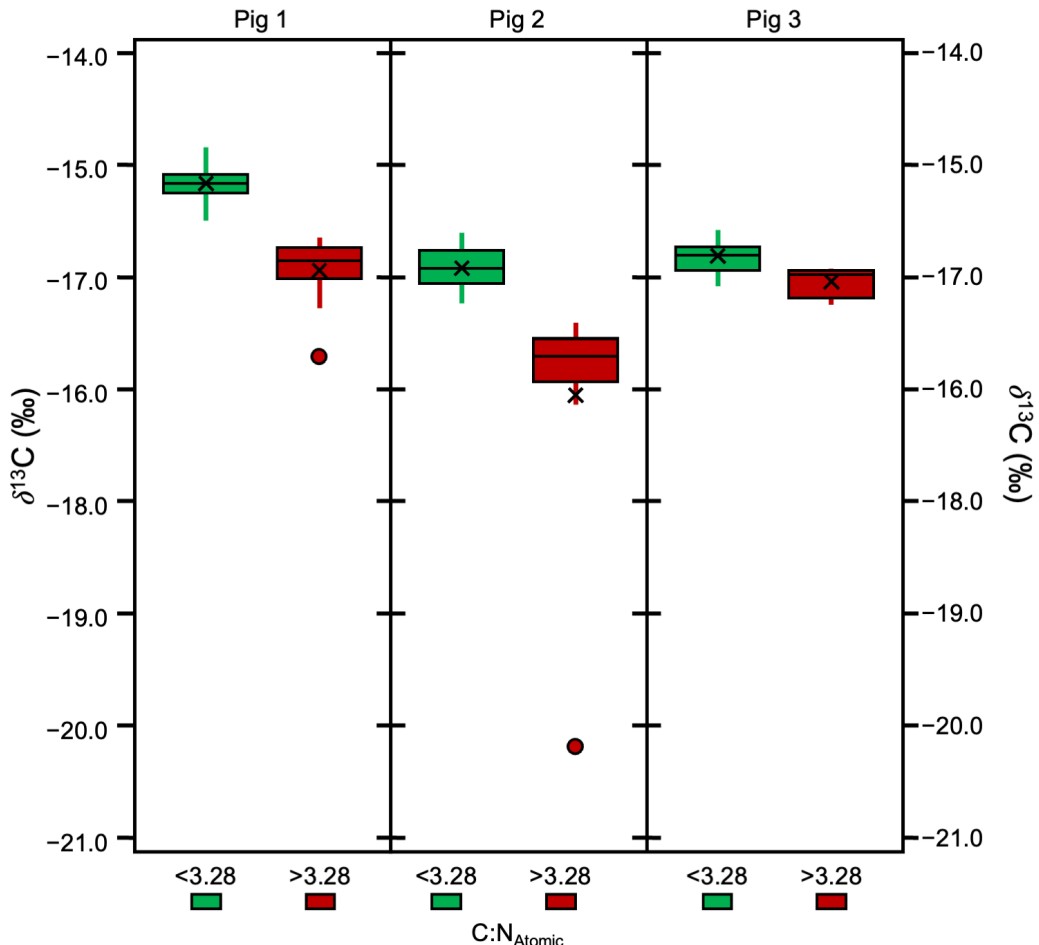

**Figure 5** Boxplot visualizing $\delta^{13}$C in samples with C:N$_{Atomic}$ ratios above and below 3.28.

62/LE*A test, all refluxed samples featured a lower standard deviation of wt% C values than those not subjected to refluxing. There were significant differences in wt% C among the treatments ($H$ [4] = 71.83, $p$ = 9.23E−15, $\varepsilon^2$ = 0.54; Fig. 6A). Dunn's *post-hoc* revealed that the differences were found between 65/LE and all other treatments (Table S6). No differences were found between the other treatments.

There were also significant differences in wt% N among treatments ($H$ [4] = 103.2, $p$ = 2.018E−21, $\varepsilon^2$ = 0.77; Fig. 6B). As with the wt% C, the significant differences were between 65/LE and all other treatments, but no other significant differences were found (Table S7). The wt% N of 0/LE had the highest standard deviation and largest range, and 65/LE had the lowest standard deviation and range (Table S2). 62/LE and 58/LE yielded nearly identical wt% N results.

## Variability in $\delta^{13}$C and $\delta^{15}$N

All refluxing treatments produced consistently low variability in isotopic compositions relative to the non-refluxed treatment (0/LE; Table S9). The combined standard deviation

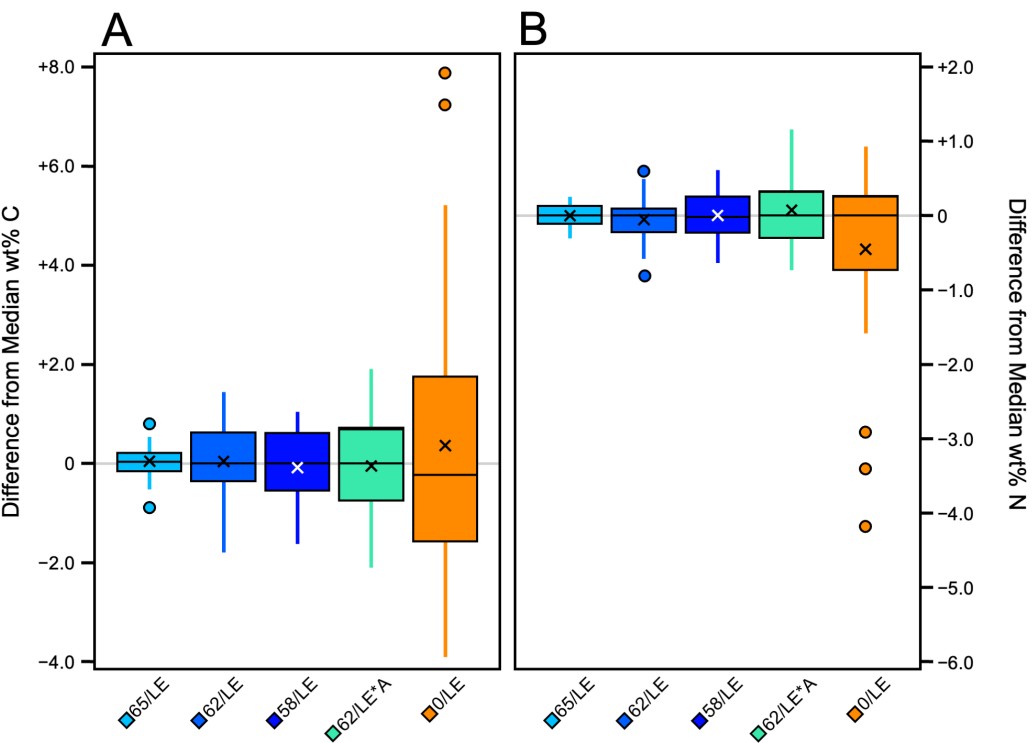

**Figure 6  Boxplot visualizing difference from distal limb median across treatments.** (A) wt% C. (B) wt% N.

of $\delta^{13}$C suggests that refluxing skin at any of the temperatures tested in this study results in more consistent $\delta^{13}$C values than non-refluxed skin. Furthermore, batches that displayed lower $\delta^{13}$C standard deviations tended to also have lower atomic C:N ratios (Fig. 7). The 0/LE samples had both higher $\delta^{13}$C standard deviations and higher atomic C:N ratios.

## Comparison of refluxed collagen and associated insoluble residue

We compared the refluxed collagen (the material that dissolved in solution during the refluxing step) *vs.* the insoluble residue (the solid material that remained insoluble after the refluxing step). The data summarized in Table S10 reflects the mean isotopic and elemental compositions, and C:N$_{Atomic}$ across collagen and residue produced by all refluxing treatments (65/LE, 62/LE, 58/LE, 62/LE*A). There were significant differences in all isotopic and elemental compositions between the refluxed material and the insoluble residue (Table S10). Refluxed collagen is characterized by lower variability in isotopic and elemental compositions, as well as a C:N$_{Atomic}$ range that is more consistent with the amino acid composition of skin. In comparison to refluxed collagen, the insoluble skin residue is characterized by larger, more widely distributed ranges of values across all isotopic and elemental compositions.

## NaOH treatment

$\delta^{13}$C and $\delta^{15}$N values generated using 62/LE*A treatment did not differ significantly from other reflux treatments, but this treatment did differ significantly from the non-refluxed

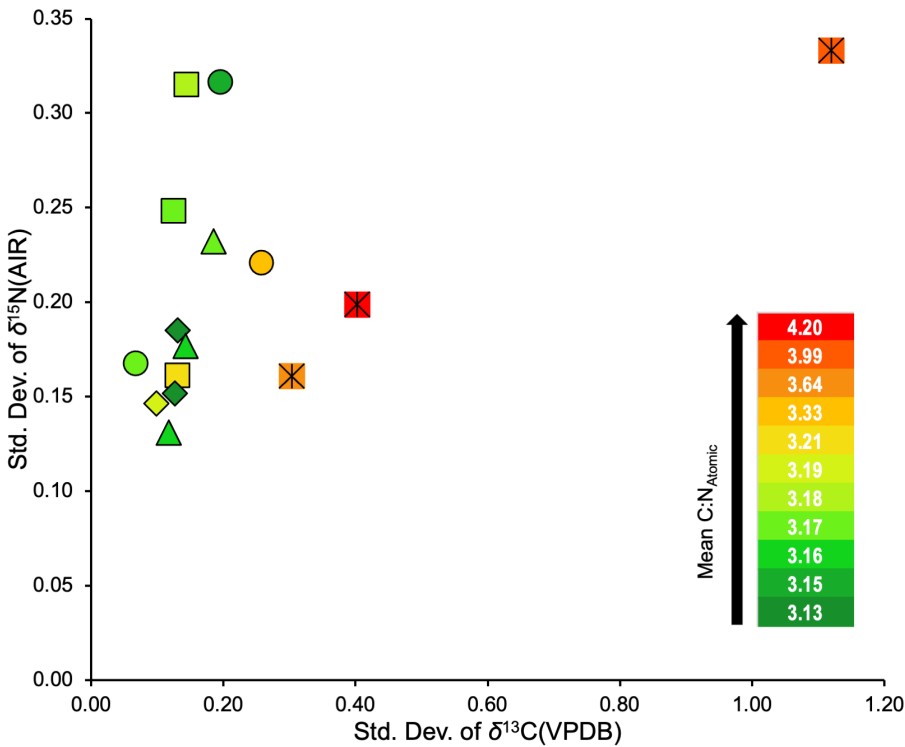

**Figure 7  Standard deviations of $\delta^{13}C$ and $\delta^{15}N$ for the three distal limbs and five treatments.** Square = 65/LE, diamond = 62/LE, triangle = 58/LE, circle = 62/LE/NaOH, crossed square = 0/LE.

treatment (Tables S3 and S4). 62/LE*A displayed standard deviations of $\delta^{13}C$ that were slightly higher than other reflux treatments, while the standard deviations of $\delta^{15}N$ were greater than all other treatments in two individuals (Table S9). Similar trends were identified in wt% C and wt% N, with 62/LE*A displaying standard deviations higher than other treatments (Tables S7–S9). The C:$N_{Atomic}$ for 62/LE*A was also higher with more variability than other reflux treatments but was still lower than 0/LE.

## 0/0 treatment

Non-lipid extracted, non-refluxed skin (0/0) from each pig was analyzed for comparison with the lipid-extracted, non-refluxed (0/LE) skin samples (Fig. 8). A Mann–Whitney U test and a $t$-test, respectively revealed that the $\delta^{13}C$ in Pigs 1 and 3 were significantly different compared to the (0/LE) samples from the same individual (Pig 1: $z = 3.37$, $p = 0.0008$, $r = 0.75$, Pig 3: $t$ (19) $= 3.03$, $p = 0.007$, $d = 1.36$). Welch's $t$-tests determined that there were significant differences in $\delta^{15}N$ between 0/0 and 0/LE of all distal limbs (Pig 1 ($t$ [18] $= 4.57$, $p = 0.0009$, $d = 2.044$); Pig 2 ($t$ [18] $= 2.22$, $p = 0.04$, $d = 0.99$); Pig 3 ($t$ [18] $= 3.12$, $p = 0.01$, $d = 1.40$)). A Mann–Whitney U test also revealed that the C:$N_{Atomic}$ differed significantly between the 0/0 and 0/LE samples ($z = 4.44$, $p = 9.1748E{-}06$, $r = 0.57$; Fig. 8).

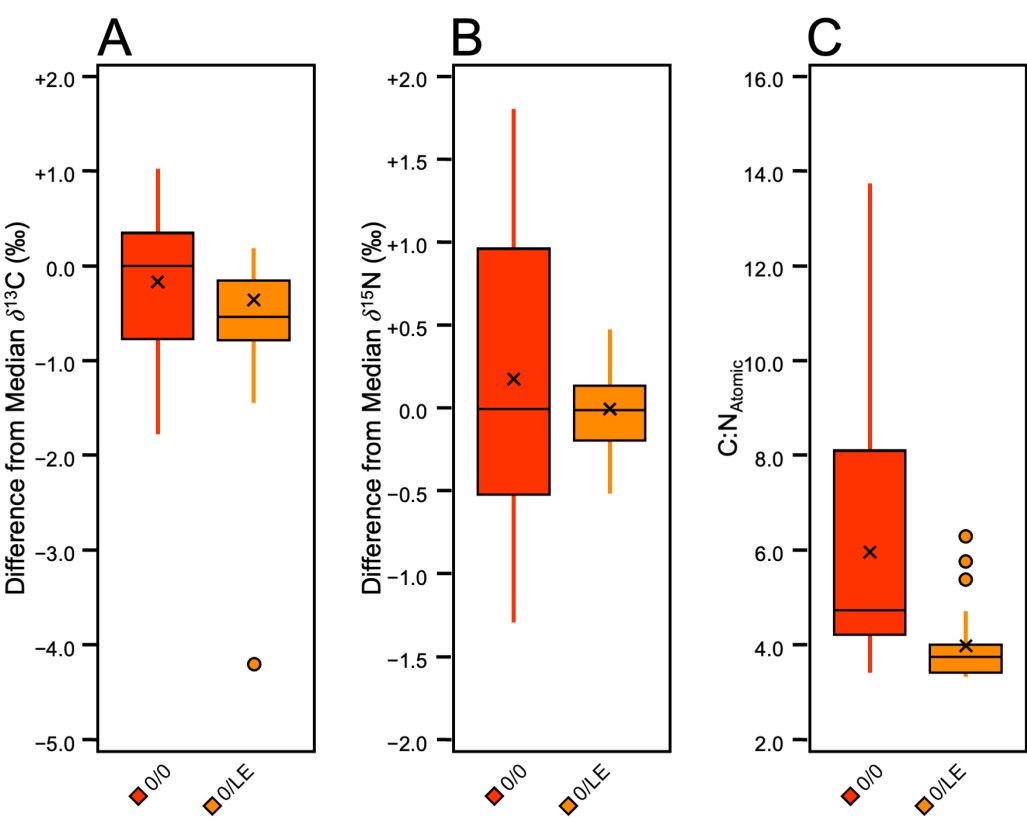

**Figure 8** **Boxplot visualizing difference from distal limb median in 0/0 and 0/LE skin samples.** (A) $\delta^{13}$C. (B) $\delta^{15}$N. C) C:N$_{Atomic}$.

## DISCUSSION

It is clear that refluxing skin results in a purer analyte that better reflects the expected elemental composition of collagen, the most abundant protein in mammalian skin. Following *Guiry & Szpak (2020)*, we consider $3.21 \pm 0.05$ to be the ideal C:N$_{Atomic}$ ratio for modern mammalian skin collagen. The elemental composition, particularly the C:N$_{Atomic}$ ratio, has been widely used for the purpose of quality control in the context of stable isotope analyses of bone collagen in ancient (*De Niro, 1985*; *Ambrose, 1990*; *van Klinken, 1999*; *Guiry & Szpak, 2021*) and modern (*Guiry & Szpak, 2020*) contexts. Samples that fall outside of a proscribed range are deemed likely to have inaccurate stable isotope measurements because the collagen is either contaminated, degraded, or some combination of the two. No comparable criteria exist for skin collagen, but since bone and skin collagen have indistinguishable elemental compositions (*Szpak, 2011*), the criteria that have been developed for modern bone collagen should be applicable to skin collagen. *Guiry & Szpak (2020)* suggested that modern mammalian bone collagen should have C:N$_{Atomic}$ between 3.00 and 3.28, with values higher than this indicating some contribution of non-collagenous material, likely lipids, NCPs (with higher C:N ratios) or some combination of these. Accordingly, we have used these data to guide our expectations for what is most likely to

reflect pure collagen that has the most homogeneous chemical composition, least likely to reflect variable contributions of different sources (*e.g.*, collagen, non-collagenous proteins, lipids).

Intact collagen is ~42% carbon and ~15% nitrogen by weight (*Guiry & Szpak, 2020*), although the %N may be higher (closer to 17.5%) for collagen that has undergone minimal deamidation of glutamine and asparagine residues (*Wilson & Szpak, 2022a*). This process occurs when collagen is acidified but it has no impact on the bulk $\delta^{13}$C and $\delta^{15}$N values of the collagen (*Wilson & Szpak, 2022b*; *Tuross, 2012*). The refluxing step includes treatment with a weak acid, but since all of the material that is dissolved is collected, the loss of amine groups through deamidation is unlikely.

None of the lipid extracted, non-refluxed (0/LE) samples had elemental compositions consistent with pure collagen, having C:N$_{Atomic}$ ratios > 3.34, and wt% C between 42.77 and 54.6. The wt% N ranged between 10.15 and 15.13. All of the non-lipid extracted, non-refluxed (0/0) samples had C:N$_{Atomic ratios}$ > 3.43, wt% C between 46.3 and 65.45, and wt% N between 5.6 and 15.8. Of the refluxed samples (from all reflux treatments), 95% ($n = 100$) had C:N$_{Atomic}$ ratios between 3.00 and 3.28. All 65/LE and 58/LE samples had C:N$_{Atomic}$ ratios between 3.00 and 3.28 and aligned well with the expectations for mammalian skin collagen. In the 62/LE batch, all but one sample (C:N of 3.37 from Pig 3) had C:N$_{Atomic}$ ratios in the expected range. All but three 62/LE*A samples had C:N$_{Atomic}$ ratios between 3.00 and 3.28. As with the 62/LE outlier, the samples outside of the 3.00–3.28 range were from Pig 3. The higher C:N$_{Atomic}$, higher wt% C, and lower $\delta^{13}$C in the outliers suggests presence of lipids or less likely, NCPs. The samples (including both refluxed collagen and whole skin) with C:N$_{Atomic}$ ratios greater than 3.28 had lower $\delta^{13}$C values than the samples with C:N$_{Atomic}$ ratios below 3.28, and the differences were significant in all three distal limbs. Our results suggest that a range of 3.0–3.28 for C:N$_{Atomic}$ is a useful measure of quality control for skin collagen.

Although there were statistically significant ($\alpha = 0.05$) differences in $\delta^{13}$C values among the refluxed batches, the means varied by <0.3‰. Furthermore, although the mean $\delta^{15}$N in the 58/LE batch from Pig 3 was significantly higher than the 65/LE and 62/LE*A samples from the same distal limb, the difference between means varied by <0.2‰. The greatest differences in isotopic compositions were found between the non-refluxed skin and the refluxed collagen. This, however, does not mean that the refluxing step altered the isotopic compositions. Instead, it suggests that the refluxing step produced material with a purer chemical composition. The non-refluxed skin had more three or more times the amount of variation in $\delta^{13}$C values than the refluxed collagen, on average. Given that the non-refluxed skin also had C:N ratios higher than expected of pure collagen, we believe that refluxing skin removed residual lipids, NCPs, or some combination of the two, thus resulting in more accurate $\delta^{13}$C values. Different biomolecules are synthesized *via* distinct pathways, each associated with their own characteristic isotopic fractionations (*Macko et al., 1986*). Accordingly, isolating as pure a substance as possible provides a better means with which to leverage the capacity of stable isotopes to act as tracers of biological, chemical, and geological processes. Individual amino acids, for example, can better trace trophic position or the isotopic composition at the base of the food web than can bulk protein samples

(*Popp et al., 2007*). Following this logic, because the inclusion of a refluxing step in the preparation of skin collagen produces a much purer analyte with respect to its elemental composition, this technique should improve the interpretive potential of stable isotope measurements.

To ensure that researchers obtain the most homogeneous, reliable, and accurate analyte possible, we recommend both lipid-extracting and refluxing skin. We do not, however, have sufficient evidence that a particular refluxing temperature produces optimal results. Based on studies of bone collagen, a temperature of 58 °C was recommended by *Brown et al. (1988)*, with higher temperatures producing lower collagen yields due to the cleavage of peptide bonds. It is important to note, however, that this collagen was subjected to centrifugal filtration, isolating the >30 kDa fraction and presumably, without this filtration step, the amount of collagen retained would not be impacted by refluxing temperature. The collagen from the 65/LE treatment adhered most closely to the expected elemental compositions of mammalian collagen, but the 62/LE treatment had the most consistent stable isotopic compositions (based on the combined standard deviations in Fig. 8). Significant differences in $C:N_{Atomic}$ were found between the 62/LE and 58/LE, and the 62/LE and 62/LE/*A treatments, but $\delta^{13}C$ values were consistent, and there is minimal evidence that the $\delta^{15}N$ values were different among the 62/LE, 58/LE, and 62/LE*A treatments. It is likely that refluxing temperatures across a relatively broad range will yield relatively pure collagen extracts, providing a minimum temperature of 58 °C at a pH of 3 (*Brown et al., 1988*) is met to break the hydrogen bonds holding together the collagen monomers.

The purpose of 62/LE*A treatment was to evaluate whether an additional treatment prior to refluxing would assist in the removal of NCPs, resulting in a purer analyte. Both fresh, modern skin, and preserved archaeological skin have occasionally been subjected to an acidification step with HCl, or acid–base-acid treatments to remove non-collagenous compounds (*Brock, 2013*; *Doherty et al., 2021*; *Davis et al., 2024*; *Doherty et al., 2022*). We did not examine any pre-refluxing acidification steps as we suspected this would only cause collagen deamidation, as discussed above. We were, however, curious whether treatment with NaOH prior to refluxing could remove NCPs with higher C:N ratios than collagen, such as elastin (C:N of 3.6–3.7) and keratin (C:N of 3.63–3.83) (*Connolly & Wu, 2024*; *Gillespie, 1990*; *Hendriks, Tarttelin & Moughan, 1998*; *Lindley, 1977*; *Miyahara, Shiozawa & Murai, 1978*; *Rothberg et al., 1965*; *Samata & Matsuda, 1988*; *Yu et al., 1993*). Contrary to our hypothesis, the mean $C:N_{Atomic}$ of the 62/LE samples was lower than in the 62/LE*A samples.

Solubilizing keratin and elastin generally requires a strong, high temperature alkali treatment (*Zhang et al., 2015*; *Halabi & Mecham, 2018*). It is therefore highly unlikely that a 30 min treatment with 0.1 M NaOH at room temperature would have been able to remove keratin or elastin, nor would it have damaged the structure of either protein so that they would be solubilized and integrated into the lyophilized collagen. Similar treatments with 0.1 or 0.125 M NaOH are regularly used in the preparation of demineralized ancient bone collagen, with minimal impact on the protein (*Szpak, Krippner & Richards, 2017*). The samples were rinsed with Type I water after the NaOH treatment, but it is possible that

some residual NaOH remained in the skin, thus raising the pH of the refluxing solution and making it less effective in solubilizing the collagen. If this happened, however, it would likely only reduce the collagen yields, and not preferentially solubilize carbon-bearing amino acids, thus resulting in higher C:N$_{Atomic}$ ratios. It is also possible that the greater variability in C:N$_{Atomic}$ ratios in the 62/LE*A treatment (compared to 62/LE) is a result of residual lipids in the refluxed collagen, but it is unlikely that the chloroform:methanol treatment was consistently less effective in one of the subsamples. We are not able to confidently explain why the 62/LE*A treatment showed greater variability in elemental composition than the other reflux treatments. Given small sample size and the lack of significant differences in the $\delta^{13}$C and $\delta^{15}$N values between the two treatments, we hesitate to attribute the higher C:N$_{Atomic}$ ratio in the 62/LE*A batch to the NaOH treatment, and reiterate that this treatment still resulted in a mean C:N$_{Atomic}$ ratio that was within the range recommended by *Guiry & Szpak (2020)* for intact collagen.

In stable isotope analysis, it is important to consider sample quality, and for biological materials, researchers should strive to isolate as pure a compound as possible. Ideally, quality control criteria that can identify samples that may be contaminated or degraded provide an excellent means of ensuring isotopic variation is caused by real biological variation and not sample preparation. Previous studies of modern (non-archaeological) skin have typically either reported C:N$_{Atomic}$ ratios but not stated a range considered indicative of good quality data (*Ryan et al., 2012*; *Kiszka et al., 2010*; *Miles, Gibbon & Hayden, 2025*), compared C:N$_{Atomic}$ ratios against a range for archaeological collagen (*Davis et al., 2024*), or have not reported C:N$_{Atomic}$ ratios (*Alves-Stanley & Worthy, 2009*; *Giménez et al., 2016*; *Todd et al., 2009*; *Todd et al., 1997*; *Borrell et al., 2018*; *Gelippi et al., 2023*; *Arregui et al., 2017*). It is common practice in archaeological skin stable isotope studies to report C:N$_{Atomic}$ (*Cockitt, Lamb & Metcalfe, 2020*; *Doherty et al., 2021*; *Davis et al., 2024*; *Finucane, 2007*; *Hyland, Millaire & Szpak, 2021*; *Doherty et al., 2022*), which is generally compared against the range recommended by *De Niro (1985)* that was developed specifically for archaeological bone collagen. Ancient and modern collagen are subject to variable concerns related to contamination and degradation and as a result, we use the range recommended by *Guiry & Szpak (2020)*, which was developed specifically for modern collagen. Similar to the findings of *Davis et al. (2024)*, our results demonstrate the importance of considering elemental data to identify samples that may be influenced by the presence of NCPs or lipids. As noted above, the two most probable NCPs to be in modern skin samples are elastin and keratin, both of which have higher C:N$_{Atomic}$ than collagen. Any relative differences in the isotopic composition of skin proteins are currently unknown. Lipids are carbon-rich and consistently depleted in $^{13}$C, which increases the %C and lower the $\delta^{13}$C of a sample, respectively (*De Niro & Epstein, 1977*).

Given that the refluxed skin collagen from all treatments had elemental compositions that best approximated pure collagen, we believe that refluxing skin is a reliable way to remove NCPs, and potentially, some lipids, producing an extract that is primarily collagen. This product is visually and chemically (on the basis of elemental compositions) indistinguishable from bone collagen (Fig. 9), which is shelf stable with respect to its isotopic and elemental compositions for at least a decade. Furthermore, the refluxed skin

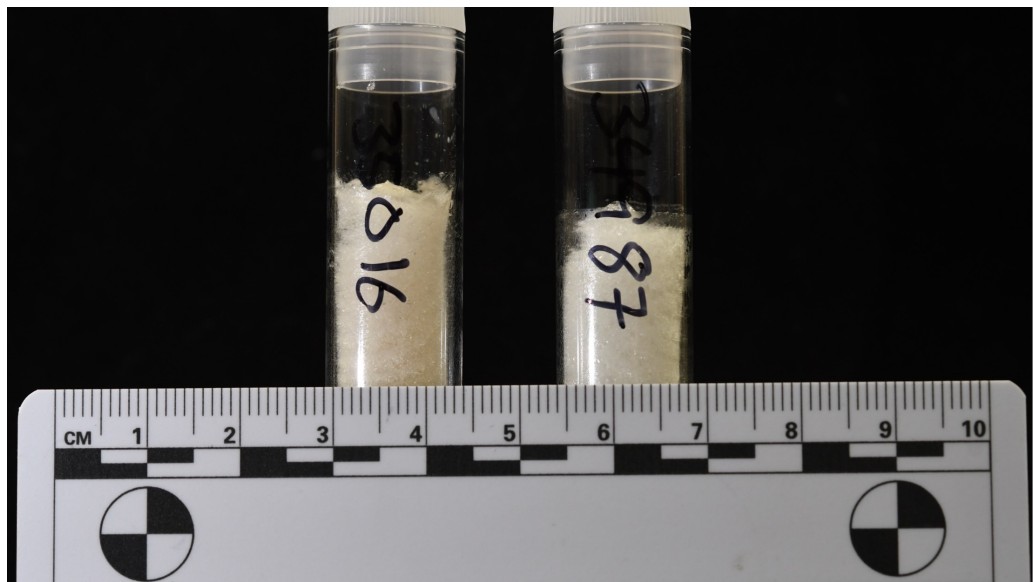

**Figure 9** Samples of refluxed bone collagen (left, 58.8 mg) and refluxed skin collagen (right, 40.9 mg) are visually indistinguishable.

collagen samples were easier to handle and could be weighed more rapidly than whole skin that was not refluxed (R Pawlowski, 2025, pers. obs. in this study). Because mammalian skin is consistent in chemical composition and structure (*Akat et al., 2022*), we believe that our results are applicable to mammalian skin in general, but future research could test whether chemical lipid extraction and refluxing skin collagen from additional species and individuals produces similar conclusions.

## CONCLUSIONS

This study investigated the impacts of three treatment variants on pig skin for stable carbon and nitrogen isotope analysis. Little has been published examining the effects of different pretreatments on modern (non-archaeological) skin, and variability in preparatory procedures between studies may impact the comparability of data. We tested three refluxing temperatures (65 °C, 62 °C, and 58 °C) and the use of NaOH, and chloroform:methanol to prepare pig skin collagen for stable isotope analysis. The refluxed collagen from all treatments more closely adhered to the expected elemental compositions of pure collagen than non-refluxed, and non-lipid extracted whole skin. Furthermore, refluxing skin has an impact on $\delta^{13}C$ values, specifically making them higher and less variable, which likely increases their accuracy if the goal is to analyze the isotopic composition of as pure a compound as possible.

Our study highlights the importance of considering how heterogeneous sample composition may alter isotopic compositions even with samples that are modern and considered to be unaltered. Non-collagenous materials can have impacts on stable isotope data, potentially impacting interpretations about an individual's diet. Given that skin

collagen reflects diet over a period of weeks to months, it is especially important that small shifts in diet can be detected through stable isotope analysis. Refluxing skin collagen is an effective way to remove other compounds that could increase the variability in $\delta^{13}C$ and $\delta^{15}N$ values.

## ACKNOWLEDGEMENTS

Matt Teeter provided technical assistance.

### Funding
Funding was provided by the Natural Sciences and Engineering Research Council of Canada (Discovery Grant 2020-04740) and Canada Research Chairs Program (950-231012). The funders had no role in study design, data collection and analysis, decision to publish, or preparation of the manuscript.

### Grant Disclosures
The following grant information was disclosed by the authors:
Natural Sciences and Engineering Research Council of Canada: 2020-04740.
Canada Research Chairs Program: 950-231012.

### Competing Interests
The authors declare there are no competing interests.

### Author Contributions

- Alexandra A.Y. Derian conceived and designed the experiments, performed the experiments, analyzed the data, prepared figures and/or tables, authored or reviewed drafts of the article, and approved the final draft.
- Ryan Pawlowski conceived and designed the experiments, performed the experiments, analyzed the data, authored or reviewed drafts of the article, and approved the final draft.
- Paul Szpak conceived and designed the experiments, prepared figures and/or tables, authored or reviewed drafts of the article, and approved the final draft.

### Data Availability
All individual isotopic and elemental compositions are available in Table S1.

### Supplemental Information
Supplemental information for this article can be found online at http://dx.doi.org/10.7717/peerj.20152#supplemental-information.

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
