# Peer review of "Solubilization of skin collagen improves the accuracy and reliability of stable isotope measurements"

_PeerJ, doi:10.7717/peerj.20152_

## Round 0.1 · original submission · Major Revisions

Reviewer 1 ·

Basic reporting

The manuscript by Derian et al. investigates the impact of three treatment variants on the stable isotope and elemental composition of pig skin using a Eurovector 3300 elemental analyzer coupled to a Nu Horizon isotope ratio mass spectrometer. The study addresses an interesting topic with potential implications for sample preparation in isotope analysis. However, several aspects of the manuscript require clarification and revision to improve its scientific rigor and clarity.

Experimental design

1. The Introduction should be revised to clearly define the research gap and justify the need for this study. Please state the primary objective of this study. Please highlight the novelty and contribution of the work.

2. The Methods section should be divided into clear sub-sections for better readability.

3. The authors should justify the selection of the specific temperatures (65, 62, and 58 °C) used in the treatments. Would a broader range provide more insight into thermal effects? The authors should discuss whether these temperatures reflect practical, industrial, or experimental relevance.

Validity of the findings

4. The figures are generally well presented and informative. However, several figures contain heavy outliers. The authors should discuss possible sources of these outliers and clarify whether they reflect true biological variation or potential experimental error.

5. Lines 168 – 175: Please cite relevant references to support the statement that skin samples are isotopically homogeneous.

6. Section 3.6: The authors should speculate on why NaOH treatment increases variability.

7. The authors mention the cutoff value of 3.28 (lines 206 – 207). Please clarify whether this cutoff is universally used and include more discussion on how lipid contamination affects isotopic data.

Reviewer 2 ·

Basic reporting

Overall, this is an interesting article about testing methods for stable isotope extraction and analysis that increases our knowledge of different protocols published. Tests like these strengthen protocols and keep us updated on the analytical advances.

Below, I present a series of questions, suggestions, and comments on the text to try to clarify some issues.

In the abstract is not necessary to explain the results. Maybe it would be good to develop the previous research in skin and tissue stable isotopes analysis and their application in scientific studies. Include the aim of the research and the samples used for the protocol test.

Suggestion: delete lines 28 to 30 in the abstract

Line 33-34: cite some examples and chronology “Has been used since …” More info about the types of tissues and organic material used for stable isotope analysis.

Line 37 also has a significant importance for the study of health and pathologies.

Line 47: such as muscle, bone, collagen, teeth, hair, etc
Maybe it would be good to differentiate between human and animal skins.

Line 57: First time said that it is pig skin, must be included before in the text
The introduction should include the aim of the paper and why it is important for the research

Experimental design

Methods:
Explain why you want to test the protocols with pig skin instead of another animal.
Why only 3 pig limbs? Are they right or left side? It’s important to know if you're sampling the same individual.
Did you know the diet of the pig? It would be good to know how they feed them, and also the quality of the water that they drink.
The age of the pig should be included as well as the health status, because these variants can interfere with subsequent results.
Line 116: Include the weight of the bone sample processing. Which part of the bone was sampled? Epiphysis or diaphysis?
Lines 111 and 120. Did you use filters?
Line 315-317: It would be good to say that more testers are needed in different bones and animal skin to have a wide range of results to compare

Validity of the findings

Line 167: Results:
It could be recommended to include a scatter plot with the results of isotopic analysis to see in a graphic way how the results from different protocols in the same samples change (or not). Two graphics, one with the skin samples and another with the bone samples
How much collagen was extracted from the different protocols?

Line 358. Conclusions
Conclusions should be longer and present the overall vision of the article.

Additional comments

Supplementary tables from 5 to 10 can be in the same Excel file with 5 different sheets.

Figure 9: Include and scale in the photo, and say how much collagen was extracted in both tubes.

Reviewer 3 ·

Basic reporting

-

Experimental design

-

Validity of the findings

-

Additional comments

The manuscript describes an empirical study of chemical pretreatments of skin for stable C and N isotope analysis, testing which approaches produce the most consistent results from sample to sample, run to run, and also whether the resulting concentrations of C and N are faithful in terms of what the theoretical C:N ratio of collagen should be. This paper should not be controversial, and a reader might be surprised it's a needed study at this late stage in isotope studies. But indeed, a review of the literature reveals that, in fact, there isn't much literature to work from. So this study, as straightforward and workmanlike as it is, does provide an empirical benchmark for others to work from.

In terms of revisions, I would just point the authors to Davis et al. 2024 Experimental Observations on Processing Leather, Skin, and Parchment for Radiocarbon Dating. Radiocarbon 66 (5):1064-1086. doi: 10.1017/RDC.2023.88. Much of the same ground is covered in that paper, focusing on radiocarbon per se, but I think considering the modes of pretreatment involved in more archaeological material and the challenges involved would help frame the scope of the present manuscript. That is, parchment, hide, leather, and skin all represent slightly different materials beyond the collagen, where perhaps using mainly modern unworked pig skin may not represent the full range of threats comprised by an archaeological specimen (or even a museum biological specimen). Some comments on the Davis paper would be useful, but I think the paper is fine for publication with just that bit of revision.

---

## Round 0.2 · accepted · Accept

Dear Dr. Derian, I congratulate you on the acceptance of this article for publication.

Reviewer 1 ·

Basic reporting

-

Experimental design

-

Validity of the findings

-

Reviewer 2 ·

Basic reporting

The article has been modified with the comments proposed in the first revision, which have enriched the article.

Experimental design

The experimental part is clear and precise, with good explanations of the protocol used and the results obtained.

Validity of the findings

-

Additional comments

Congratulations on the article. I hope it's the beginning of research that will yield results that can be used in various fields of scientific research.

Reviewer 3 ·

Basic reporting

-

Experimental design

-

Validity of the findings

-

Additional comments

The authors have responded to my suggestions from the first round of review.